# How to Train Your MAML
# to Excel in Few-Shot Classification

**Han-Jia Ye**
State Key Laboratory for Novel Software Technology, Nanjing University

**Wei-Lun Chao**
The Ohio State University

## Abstract

Model-agnostic meta-learning (MAML) is arguably one of the most popular meta-learning algorithms nowadays. Nevertheless, its performance on few-shot classification is far behind many recent algorithms dedicated to the problem. In this paper, we point out several key facets of how to train MAML to excel in few-shot classification. First, we find that MAML needs a large number of gradient steps in its inner loop update, which contradicts its common usage in few-shot classification. Second, we find that MAML is sensitive to the class label assignments during meta-testing. Concretely, MAML meta-trains the initialization of an $N$-way classifier. These $N$ ways, during meta-testing, then have "$N!$" different permutations to be paired with a few-shot task of $N$ novel classes. We find that these permutations lead to a huge variance of accuracy, making MAML unstable in few-shot classification. Third, we investigate several approaches to make MAML permutation-invariant, among which meta-training a *single vector to initialize all the $N$ weight vectors in the classification head* performs the best. On benchmark datasets like *Mini*ImageNet and *Tiered*ImageNet, our approach, which we name UNICORN-MAML, performs on a par with or even outperforms many recent few-shot classification algorithms, *without sacrificing MAML's simplicity*.

## 1 Introduction

Meta-learning is a sub-field of machine learning which attempts to search for the best learning strategy as the learning experiences increases (Thrun & Pratt, 2012; Lemke et al., 2015). Recent years have witnessed an abundance of new approaches on meta-learning (Vanschoren, 2018; Hospedales et al., 2020), among which model-agnostic meta-learning (MAML) (Finn et al., 2017; Finn, 2018) is one of the most popular algorithms, owing to its "model-agnostic" nature to incorporate different model architectures and its principled formulation to be applied to different problems. Concretely, MAML aims to learn a good *model initialization* (through the outer loop optimization), which can then be quickly adapted to novel tasks given few examples (through the inner loop optimization).

However, in few-shot classification (Vinyals et al., 2016; Snell et al., 2017) which many meta-learning algorithms are dedicated to, MAML's performance has been shown to fall far behind (Wang et al., 2019; Chen et al., 2019; Triantafillou et al., 2020).

In this paper, we take a closer look at MAML on few-shot classification. The standard setup involves two phases, meta-training and meta-testing, in which MAML learns the model initialization during meta-training and applies it during meta-testing. In both phases, MAML receives multiple $N$-way $K$-shot tasks. Each task is an $N$-class classification problem provided with $K$ labeled support examples per class. After the (temporary) inner loop optimization using the support examples, the updated model from the initialization is then evaluated on the query examples of the same $N$ classes. The loss calculated on the query examples during mete-training is used to optimize the meta-parameters (*i.e.*, the model initialization) through the outer loop. *For consistency, we mainly study the scenario where meta-training and meta-testing use the same number of gradient steps in the inner loop optimization.*

More specifically, what MAML learns for few-shot classification is the initialization of an $N$-class classifier. Without loss of generality, we denote a classifier by $\hat{y} = \arg\max_{c \in [N]} \boldsymbol{w}_c^\top f_{\boldsymbol{\phi}}(\boldsymbol{x})$, where $f_{\boldsymbol{\phi}}$ is the feature extractor on an example $\boldsymbol{x}$; $\{\boldsymbol{w}_c\}_{c=1}^N$ are the weight vectors in the linear classification head. We use $\boldsymbol{\theta}$ to represent the collection of meta-parameters $\{\boldsymbol{\phi}, \boldsymbol{w}_1, \cdots, \boldsymbol{w}_N\}$.

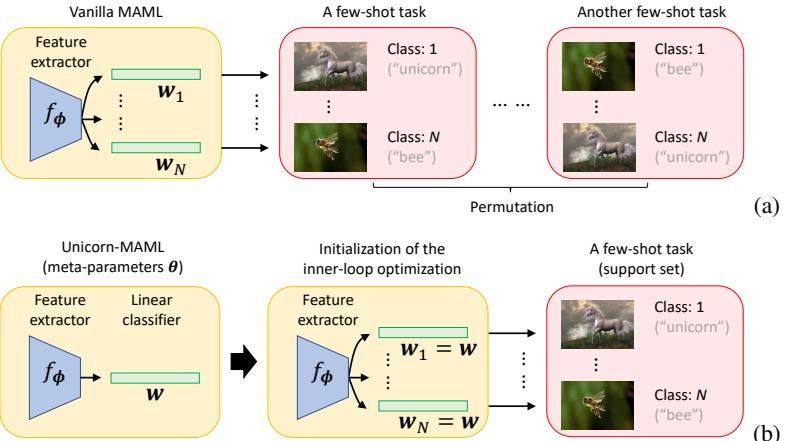

Figure 1: **The problem of permutations in label assignments, and the illustration of UNICORN-MAML.** (a) A vanilla MAML learns the initialization of $\phi$ and $\{w_c\}_{c=1}^N$ (*i.e.*, the $N$ weight vectors). Each of $\{w_c\}_{c=1}^N$ is paired with the corresponding class label $c \in [N]$ of a few-shot task. A few-shot task, however, may consist of the same set of semantic classes but in different permutations of class label assignments, leading to a larger variance in meta-testing accuracy. (b) In contrast, our **UNICORN-MAML**, besides learning $\phi$, learns only a single weight vector $w$ and uses it to initialize all the $N$ weight vectors $\{w_c\}_{c=1}^N$ at the beginning of the inner loop. That is, UNICORN-MAML directly forces the learned model initialization to be permutation-invariant.

Our first observation is that **MAML needs a large number of gradient steps in the inner loop**. For example, on *Mini*ImageNet (Vinyals et al., 2016) and *Tiered*ImageNet (Ren et al., 2018a), MAML's accuracy improves along with the increased number of gradient steps and achieves the highest around $15 \sim 20$ steps, which are much larger than the conventional usage of MAML (Antoniou et al., 2019). We attribute this to the behavior of the model initialization learned from mutually-exclusive tasks (Yin et al., 2020), which, without any further inner loop optimization, performs at the chance level (*i.e.*, $\frac{100}{N}\%$ accuracy) on query examples, not only for the meta-testing tasks but also for the meta-training tasks. In other words, the initialized model needs many gradient steps to attain a high accuracy.

Our second observation is that **MAML is sensitive to the permutations of class label assignments during meta-testing**. Concretely, when an $N$-way $K$-shot task arrives, MAML pairs the learned initialization of $w_c$ with the corresponding class label $c \in [N]$ of that task. The issue resides in the "meaning" of $c$ in a task. In the standard setup, each $N$-way task is created by drawing $N$ classes from a bigger pool of semantic classes (*e.g.*, "dog", "cat", "bird", etc.), followed by a *random* label re-assignment into $c \in \{1, \cdots, N\}$. In other words, the same set of $N$ semantic classes can be labeled totally differently into $\{1, \cdots, N\}$ and thus be paired with $\{w_c\}_{c=1}^N$ differently. Taking a five-way task for example, there are $5! = 120$ permutations to pair the same set of five semantic classes to the linear classification head. In some of them, a class "dog" may be assigned to $c = 1$; in some others, it may be assigned to $c \neq 1$. *While this randomness has been shown crucial in meta-training to help MAML prevent over-fitting (Rajendran et al., 2020; Yao et al., 2021; Yin et al., 2020), we find that it makes the meta-testing phase unstable.* Specifically, different permutations can lead to drastically different meta-testing accuracy — on average, the best permutation for each five-way one-shot task has $\sim 15\%$ higher accuracy than the worst permutation, on both datasets.

Building upon this observation, we investigate multiple approaches to **make MAML permutation-invariant**, either in the meta-testing phase alone or in both phases. We find that a simple solution — *meta-training only a single vector $w$ and using it to initialize the $N$ linear classifiers $\{w_c\}_{c=1}^N$* — performs the best. We name this approach UNICORN-MAML, as illustrated in Figure 1 (b). Concretely, at the beginning of the inner loop, UNICORN-MAML duplicates $w$ into $w_c, \forall c \in [N]$. After the inner loop optimization, the meta-gradient with respect to $\{w_c\}_{c=1}^N$ are aggregated to update $w$ in the outer loop (during meta-training). This design not only makes UNICORN-MAML permutation-invariant, but also ensures that no single model can solve all tasks at once without inner loop optimization, to prevent memorization over-fitting (Yin et al., 2020). On *Mini*ImageNet (Vinyals et al., 2016), *Tiered*ImageNet (Ren et al., 2018a), and CUB datasets (Wah et al., 2011), UNICORN-MAML performs on a par with or even outperforms many recent few-shot classification algorithms, while preserving the simplicity of MAML without adding any extra network modules or learning strategies. Our code is available at `https://github.com/Han-Jia/UNICORN-MAML`.

## 2 MAML FOR FEW-SHOT CLASSIFICATION

### 2.1 PROBLEM DEFINITION

The goal of few-shot classification is to construct a classifier using limited labeled examples. The challenge is the potential over-fitting or poor generalization problem. Following (Vinyals et al., 2016), we define a few-shot classification problem as an $N$-way $K$-shot task, which has $N$ classes and each class has $K$ labeled support examples. We denote the labeled support set by $\mathcal{S} = \{(\boldsymbol{x}_i, y_i)\}_{i=1}^{N \times K}$; each $(\boldsymbol{x}_i, y_i)$ is a pair of an input (*e.g.*, image) and a class label, where $y_i \in \{1, 2, \cdots, N\} = [N]$. The value of $K$ is small, *e.g.*, $K = 1$ or $K = 5$. To evaluate the quality of the resulting classifier, each task is associated with a query set $\mathcal{Q}$, which is composed of examples of the same $N$ classes.

The core idea of meta-learning for few-shot classification is to sample few-shot tasks $\mathcal{T} = (\mathcal{S}, \mathcal{Q})$ from a set of "base" classes, of which we have ample examples per class. Meta-learning then learns the ability of *"how to build a classifier using limited data"* from these tasks. After this **meta-training** phase, we then proceed to the **meta-testing** phase to tackle the true few-shot tasks that are composed of examples from "novel" classes. By default, the "novel" and "base" classes are disjoint. It is worth noting that the total number of "base" (and "novel") classes is usually larger than $N$ (see subsection 2.3). Thus, to construct an $N$-way $K$-shot task in each phase, one usually first samples $N$ classes from the corresponding set of classes, and randomly re-labels each sampled class by an index $c \in [N]$. This randomness results in the so-called mutually-exclusive tasks (Yin et al., 2020). In this paper, we will use base (novel) and meta-training (meta-testing) classes interchangeably.

### 2.2 MODEL-AGNOSTIC META-LEARNING (MAML)

As introduced in section 1, MAML aims to learn the initialization of an $N$-way classifier, such that when provided with the support set $\mathcal{S}$ of an $N$-way $K$-shot task, the classifier can be quickly and robustly updated to perform well on the task (*i.e.*, classify the query set $\mathcal{Q}$ well). Let us denote a classifier by $\hat{y} = h_{\boldsymbol{\theta}}(\boldsymbol{x}) = \arg\max_{c \in [N]} \boldsymbol{w}_c^\top f_{\boldsymbol{\phi}}(\boldsymbol{x})$, where $f_{\boldsymbol{\phi}}$ is the feature extractor, $\{\boldsymbol{w}_c\}_{c=1}^N$ are the weight vectors of the classification head, and $\boldsymbol{\theta} = \{\boldsymbol{\phi}, \boldsymbol{w}_1, \cdots, \boldsymbol{w}_N\}$ collects the parameters of both. MAML evaluates $h_{\boldsymbol{\theta}}$ on $\mathcal{S}$ and uses the gradient to update $\boldsymbol{\theta}$ into $\boldsymbol{\theta}'$, so that $h_{\boldsymbol{\theta}'}$ can be applied to $\mathcal{Q}$. This procedure is called the **inner loop optimization**, which usually takes $M$ gradient steps.

$$
\begin{aligned}
&\boldsymbol{\theta}' \leftarrow \boldsymbol{\theta} \\
&\text{for } m \in [M] \text{ do} \\
&\quad \boldsymbol{\theta}' = \boldsymbol{\theta}' - \alpha \nabla_{\boldsymbol{\theta}'} \mathcal{L}(\mathcal{S}, \boldsymbol{\theta}')
\end{aligned}
\tag{1}
$$

Here, $\mathcal{L}(\mathcal{S}, \boldsymbol{\theta}') = \sum_{(\boldsymbol{x}, y) \in \mathcal{S}} \ell(h_{\boldsymbol{\theta}'}(\boldsymbol{x}), y)$ is the loss computed on examples of $\mathcal{S}$ and $\alpha$ is the learning rate (or step size). The cross-entropy loss is commonly used for $\ell$. As suggested in the original MAML paper (Finn et al., 2017) and (Antoniou et al., 2019), $M$ is usually set to a small integer (*e.g.*, $\leq 5$). For ease of notation, let us denote the output $\boldsymbol{\theta}'$ after $M$ gradient steps by $\boldsymbol{\theta}' = \mathsf{InLoop}(\mathcal{S}, \boldsymbol{\theta}, M)$.

To learn the initialization $\boldsymbol{\theta}$, MAML leverages the few-shot tasks sampled from the base classes. Let us denote by $p(\mathcal{T})$ the distribution of tasks from the base classes, where each task is a pair of support and query sets $(\mathcal{S}, \mathcal{Q})$. MAML aims to minimize the following meta-training objective w.r.t. $\boldsymbol{\theta}$:

$$
\mathbb{E}_{(\mathcal{S}, \mathcal{Q}) \sim p(\mathcal{T})} \mathcal{L}(\mathcal{Q}, \boldsymbol{\theta}'_{\mathcal{S}}) = \mathbb{E}_{(\mathcal{S}, \mathcal{Q}) \sim p(\mathcal{T})} \mathcal{L}(\mathcal{Q}, \mathsf{InLoop}(\mathcal{S}, \boldsymbol{\theta}, M)).
\tag{2}
$$

Namely, MAML aims to find a shared $\boldsymbol{\theta}$ among tasks, which, after inner loop updates using $\mathcal{S}$, can lead to a small loss on the query set $\mathcal{Q}$. (We add the subscript $\mathcal{S}$ to $\boldsymbol{\theta}'$ to show that $\boldsymbol{\theta}'_{\mathcal{S}}$ depends on $\mathcal{S}$.) To optimize Equation 2, MAML applies stochastic gradient descent (SGD) but at the task level. That is, at every iteration, MAML samples a task $\mathcal{T} = (\mathcal{S}, \mathcal{Q})$ and computes the meta-gradient w.r.t. $\boldsymbol{\theta}$:

$$
\nabla_{\boldsymbol{\theta}} \mathcal{L}(\mathcal{Q}, \boldsymbol{\theta}'_{\mathcal{S}}) = \nabla_{\boldsymbol{\theta}} \mathcal{L}(\mathcal{Q}, \mathsf{InLoop}(\mathcal{S}, \boldsymbol{\theta}, M)).
\tag{3}
$$

In practice, one may sample a mini-batch of tasks and compute the mini-batch meta-gradient w.r.t. $\boldsymbol{\theta}$ to optimize $\boldsymbol{\theta}$. This SGD for $\boldsymbol{\theta}$ is known as the **outer loop optimization** for MAML. It is worth noting that calculating the gradient in Equation 3 can impose considerable computational and memory burdens because it involves a gradient through a gradient (along the inner loop but in a backward order) (Finn et al., 2017). Thus in practice, it is common to apply the first-order approximation (Finn et al., 2017; Nichol et al., 2018), *i.e.*, $\nabla_{\boldsymbol{\theta}} \mathcal{L}(\mathcal{Q}, \boldsymbol{\theta}'_{\mathcal{S}}) \approx \nabla_{\boldsymbol{\theta}'_{\mathcal{S}}} \mathcal{L}(\mathcal{Q}, \boldsymbol{\theta}'_{\mathcal{S}})$.

**For additional related work on meta-learning and few-shot learning, please see Appendix A.**

## 2.3 EXPERIMENTAL SETUP

As our paper is heavily driven by empirical observations, we first introduce the three main datasets we experiment on, the neural network architectures we use, and the implementation details.

**Dataset.** We work on ***Mini*ImageNet** (Vinyals et al., 2016), ***Tiered*ImageNet** (Ren et al., 2018a), and **CUB** datasets (Wah et al., 2011). *Mini*ImageNet contains 100 semantic classes; each has 600 images. Following (Ravi & Larochelle, 2017), the 100 classes are split into meta-training/validation/testing sets with 64/16/20 (non-overlapped) classes, respectively. That is, there are 64 base classes and 20 novel classes; the other 16 classes are used for hyper-parameter tuning. *Tiered*ImageNet (Ren et al., 2018a) has 608 semantic classes, which are split into the three sets with 351/97/160 classes, respectively. On average, each class has $\sim 1,300$ images. CUB (Wah et al., 2011) has 200 classes, which are split into the three sets with 200/50/50 classes following (Ye et al., 2020a). All images are resized to $84 \times 84$, following (Lee et al., 2019; Ye et al., 2020a).

**Training and evaluation.** During meta-training, meta-validation, and meta-testing, we sample $N$-way $K$-shot tasks from the corresponding classes and images. We follow the literature (Snell et al., 2017; Vinyals et al., 2016) to study the five-way one-shot and five-way five-shot tasks. As mentioned in subsection 2.1, every time we sample five distinct classes, we randomly assign each of them an index $c \in [N]$. During meta-testing, we follow the evaluation protocol in (Zhang et al., 2020; Rusu et al., 2019; Ye et al., 2020a) to sample $10,000$ tasks. In each task, the query set contains 15 images per class. We report the mean accuracy (in %) and the $95\%$ confidence interval.

**Model architecture.** We follow (Lee et al., 2019) to use a ResNet-12 (He et al., 2016) architecture for $f_\phi$ (cf. subsection 2.2), which has wider widths and Dropblock modules (Ghiasi et al., 2018). We note that many recent few-shot learning algorithms use this backbone. We also follow the original MAML (Finn et al., 2017) to use a 4-layer convolutional network (ConvNet) (Vinyals et al., 2016).

**Implementation details.** Throughout the paper, for simplicity and consistency, we use

- the first-order approximation for calculating the meta-gradient in the outer loop;
- the same number of gradient steps in the inner loop during meta-training and meta-testing;
- the weights pre-trained on the entire meta-training set to initialize $\phi$, following the recent practice (Ye et al., 2020a; Rusu et al., 2019; Qiao et al., 2018). *We note that in meta-training we still optimize this pre-trained $\phi$ in the "outer" loop to search for a better initialization for MAML.*

MAML has several hyper-parameters and we select them on the meta-validation set. Specifically, for the outer loop, we learn with at most $10,000$ tasks: we group every 100 tasks into an epoch. We apply SGD with momentum 0.9 and weight decay 0.0005. We start with an outer loop learning rate 0.002 for ConvNet and 0.001 for ResNet-12, which are decayed by 0.5 and 0.1 after every 20 epochs for ConvNet and ResNet-12, respectively. For the inner loop, we have to set the number of gradient steps $M$ and the learning rate $\alpha$ (cf. Equation 1). We provide more details in the next section.

## 3 MAML NEEDS A LARGE NUMBER OF INNER LOOP GRADIENT STEPS

We find that for MAML's inner loop, the number of gradient updates $M$ (cf. Equation 1) is usually selected within a small range close to 1, *e.g.*, $M \in [1, 5]$ (Antoniou et al., 2019). At first glance, this makes sense according to the motivation of MAML (Finn et al., 2017) — with a small number of gradient steps, the resulting model will have a good generalization performance.

In our experiment, we however observe that it is crucial to explore a larger $M$[1]. Specifically, we consider $M \in [1, 20]$ along with $\alpha \in [10^{-4}, 10^0]$. We plot the meta-testing accuracy of five-way one-shot tasks on the three datasets in Figure 2[2], using both ResNet and ConvNet backbones. We find that MAML achieves higher and much more stable accuracy (w.r.t. the learning rate) when $M$ is larger, *e.g.*, larger than 10. Specifically, for *Mini*ImageNet with ResNet, the highest accuracy $64.42\%$ is obtained with $M = 15$, higher than $62.90\%$ with $M = 5$; for *Tiered*ImageNet with ResNet, the highest accuracy $65.72\%$ is obtained with $M = 15$, higher than $59.08\%$ with $M = 5$. As will be seen in section 6, these results with a larger $M$ are already close to the state-of-the-art performance.

---

[1] We reiterate that for simplicity and consistency we apply the same $M$ in meta-training and meta-testing.

[2] We tune hyper-parameters on the meta-validation set and find that the accuracy there reflects the meta-testing accuracy well. We show the meta-testing accuracy here simply for a direct comparison to results in the literature.

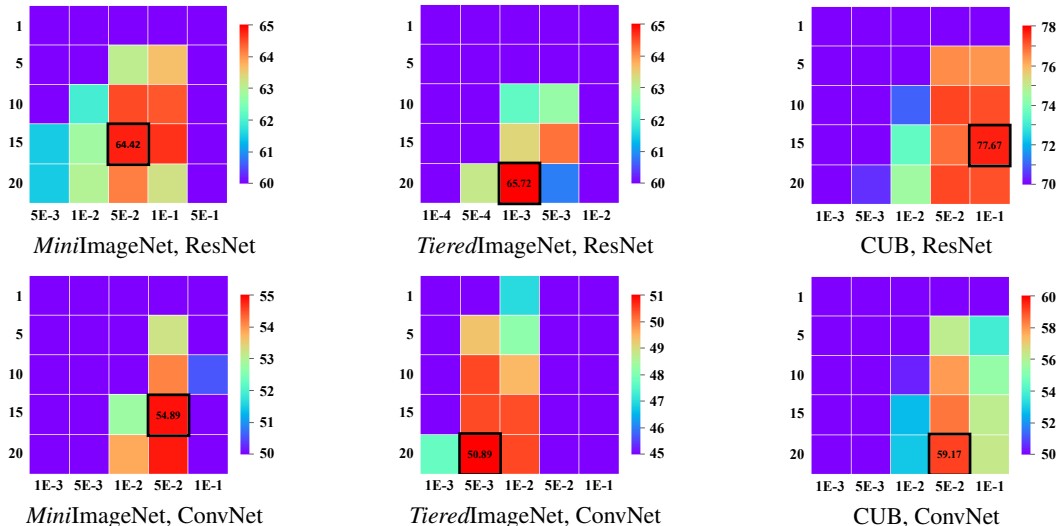

Figure 2: Heat maps of MAML's five-way one-shot accuracy on *Mini*ImageNet, *Tiered*ImageNet, and CUB w.r.t. the inner loop learning rate $\alpha$ (x-axis) and the number of inner loop updates $M$ (y-axis). For each heat map, **we set accuracy below a threshold to a fixed value for clarity**; we denote the best accuracy by a black box.

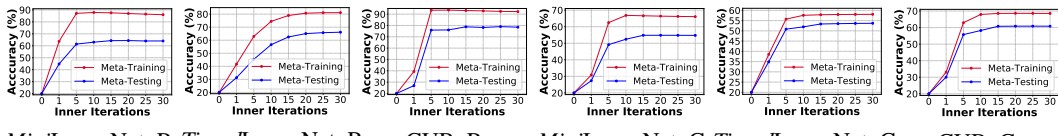

Figure 3: We plot the change of the five-way one-shot classification accuracy (on the query set), averaged over 10,000 tasks sampled from either the meta-training (red) or meta-testing classes (blue), along with the process of inner loop updates, using the best model initialization learned by MAML. C: ConvNet; R: ResNet.

To analyze why MAML needs a large $M$, we plot the change of classification accuracy along with the inner loop updates in Figure 3. Specifically, we first perform meta-training using the $M$ value selected by meta-validation, for each pair of dataset and backbone. We then analyze the learned model initialization on few-shot tasks sampled from the meta-training and meta-testing classes, by performing $0 \sim 30$ inner loop updates using the support set and reporting the accuracy on the query set. **We conduct the same experiments for five-way five-shot tasks in Appendix D.**

We have two observations. First, in general, the more inner loop updates we perform for a few-shot task, the higher the accuracy is, no matter if it is a meta-training or meta-testing task. This trend aligns with the few-shot regression study in (Finn et al., 2017). Second, before any inner loop update, the learned initialization $\boldsymbol{\theta}$ has a $\sim 20\%$ accuracy on average, *i.e.*, the accuracy by random classification. Interestingly, this is the case not only for meta-testing tasks but also for meta-training tasks, even though the learned initialization does contain the classification head $\{\boldsymbol{w}_c\}_{c=1}^N$. *This explains why a larger number of inner loop gradient steps is needed: the learned initialized model has to be updated from performing random predictions to achieving a much higher classification accuracy.*

We attribute the second observation to the *random* class label assignments in creating few-shot tasks (cf. subsection 2.1 and subsection 2.3), which make the created tasks mutually-exclusive (Yin et al., 2020) — *i.e.*, a single model cannot solve them all at once before inner loop optimization. Concretely, for a few-shot task of a specific set of $N$ semantic classes (*e.g.*, {"dog", "cat", $\cdots$, "bird"}), such a randomness can turn it into different tasks from MAML's perspective. For instance, the class "dog" may be assigned to $c = 1$ and paired with $\boldsymbol{w}_1$ at the current task, but to $c = 2$ and paired with $\boldsymbol{w}_2$ when it is sampled again. *For a five-way task, the same set of five semantic classes can be assigned to $\{1, \cdots, 5\}$ via 120 (i.e., 5!) different permutations.* As a result, if we directly apply the learned initialization of MAML without inner loop updates, the accuracy on few-shot tasks of the same set of semantic classes (but in different permutations) can cancels each other out. (Please see subsection C.1 for details.) Besides, since the randomness occurs also in meta-training, each $\boldsymbol{w}_c$ will be discouraged to learn specific knowledge towards any semantic class (Rajendran et al., 2020; Yao et al., 2021; Yin et al., 2020), hence producing an accuracy at the chance level even on meta-training tasks.

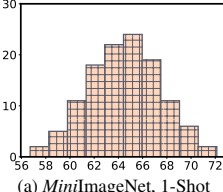 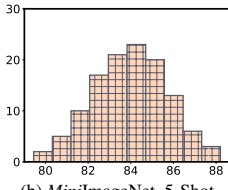 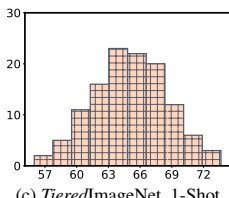 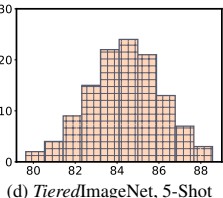

(a) *Mini*ImageNet, 1-Shot  (b) *Mini*ImageNet, 5-Shot  (c) *Tiered*ImageNet, 1-Shot  (d) *Tiered*ImageNet, 5-Shot

Figure 4: The histogram of the 120 meta-testing accuracy (averaged over 2, 000 tasks), each corresponds to a specific position in the sorted list of each task's accuracy among 120 permutations. The x-axis corresponds to accuracy (range); the y-axis corresponds to counts. The backbone is ResNet-12.

---

**Algorithm 1:** Evaluation of the effect of class label permutations on meta-testing tasks.

---

**Given** the learned initialization $\boldsymbol{\theta}$ by MAML
**for** $t \in \{1, \cdots, 2000\}$ **do**
    **Sample** a meta-testing task $\mathcal{T} = (\mathcal{S}, \mathcal{Q})$ and initialize an accuracy vector $\boldsymbol{a}_t \in \mathbb{R}^{120}$
    **for** $p \in \{1, \cdots, 120\}$ **do**
        **Shuffle** the class labels with a specific permutation $\pi : [N] \mapsto [N]$; *i.e.*, $(\mathcal{S}, \mathcal{Q})$ becomes $(\mathcal{S}_\pi, \mathcal{Q}_\pi)$
        **Update** $\boldsymbol{\theta}$ to get $\boldsymbol{\theta}' = \mathsf{InLoop}(\mathcal{S}_\pi, \boldsymbol{\theta}, M)$, evaluate $\boldsymbol{\theta}'$ on $\mathcal{Q}_\pi$, and record the accuracy into $\boldsymbol{a}_t[p]$
    **end**
    **Sort** values in the accuracy vector $\boldsymbol{a}_t$ in the descending order and get $\boldsymbol{a}_t^{(\text{sort})}$
**end**
**Average** over 2000 tasks by $\frac{1}{2000} \sum_t \boldsymbol{a}_t^{(\text{sort})}$

---

**Related work.** Several following-up works of MAML, *e.g.*, (Hsu et al., 2019), use different numbers of inner loop steps in meta-training (*e.g.*, $M = 1 \sim 5$) and meta-testing (*e.g.*, $M = 20 \sim 50$). We make $M$ equal in the two phases for consistency, and provide a detailed analysis of why MAML needs a large $M$ value. On large-scale tasks beyond few-shot learning, Shin et al. (2021) also found the necessity of a larger number of inner loop steps, but from a different perspective than ours.

## 4 MAML is Sensitive to The Label Permutations in Meta-Testing

The randomness in class label assignments raises an interesting question: *do different permutations result in different meta-testing accuracy after inner loop updates?* More specifically, if $\{\boldsymbol{w}_c\}_{c=1}^N$ are paired with the $N$ classes differently, will the updated model after the inner loop perform differently?

To answer this question, we conduct a detailed experiment: Algorithm 1 summarizes the procedure. We focus on **five-way one/five-shot** tasks on *Mini*ImageNet and *Tiered*ImageNet, using the ResNet backbone. For each task type and dataset combination, we first meta-train the model initialization using MAML, and then evaluate the learned initialized on 2, 000 meta-testing tasks. For each task, there are 120 permutations; each permutation, after the inner loop, would likely lead to a different model and query set accuracy. *We sort the 120 accuracy for each task, and take average for each position in the sorted list over 2, 000 tasks.* This results in 120 averaged accuracy, each for a specific position in the sorted list. Specifically, the highest accuracy corresponds to the case that each task cherry-picks its best permutation according to the query set accuracy after inner loop optimization.

We show the histogram of the 120 average meta-testing accuracy in Figure 4. There exists a huge variance. Specifically, the best permutation can be $15\%/8\%$ higher than the worst in one/five-shot tasks. The best permutation is also much higher than vanilla MAML's results (from section 3), which are 64.42%/83.44%/65.72%/84.37%, corresponding to the four sub-figures from left to right in Figure 4. What's more, the best permutation can easily achieve state-of-the-art accuracy (see section 6).

Of course, so far we find the best permutation via cherry-picking — by looking at the meta-testing accuracy — so it is like an upper bound. However, if we can find the best permutation without looking at the query sets' labels or make MAML permutation-invariant, we may practically improve MAML.

**Related work.** We note that we investigate the effect of the permutations of class label assignments differently from (Rajendran et al., 2020; Yao et al., 2021; Yin et al., 2020). There, the permutations lead to mutually-exclusive meta-training tasks, which help prevent MAML from over-fitting. Here, we look at the meta-testing tasks, and permutations result in a huge variance of meta-testing accuracy.

Table 1: The meta-testing accuracy over 2,000 tasks given different permutation selection strategies.

| | *Mini*ImageNet | | *Tiered*ImageNet | |
|---|---|---|---|---|
| Select the permutation by | 1-Shot | 5-Shot | 1-Shot | 5-Shot |
| None | $64.42 \pm 0.20$ | $83.44 \pm 0.13$ | $65.72 \pm 0.20$ | $84.37 \pm 0.16$ |
| Initial Support Acc | $64.42 \pm 0.20$ | $83.95 \pm 0.13$ | $65.06 \pm 0.20$ | $84.32 \pm 0.16$ |
| Initial Support Loss | $64.42 \pm 0.20$ | $83.91 \pm 0.13$ | $65.42 \pm 0.20$ | $84.23 \pm 0.16$ |
| Updated Support Acc | $64.42 \pm 0.20$ | $83.95 \pm 0.13$ | $65.01 \pm 0.20$ | $84.37 \pm 0.16$ |
| Updated Support Loss | $64.67 \pm 0.20$ | $84.05 \pm 0.13$ | $65.43 \pm 0.20$ | $84.22 \pm 0.16$ |

Table 2: Ensemble over updated models of different permutations. (The confidence interval is omitted due to space limit.)

| *Mini* | Vanilla | Full | Rotated | | *Tiered* | Vanilla | Full | Rotated |
|---|---|---|---|---|---|---|---|---|
| 1-Shot | 64.42 | 65.50 | 65.37 | | 1-Shot | 65.72 | 66.68 | 66.63 |
| 5-Shot | 83.44 | 84.43 | 84.40 | | 5-Shot | 84.37 | 84.83 | 84.81 |

Table 3: We average the top-layer classifiers and expand it to $N$-way during meta-testing.

| | *Mini*ImageNet | *Tiered*ImageNet |
|---|---|---|
| 1-Shot | $64.40 \pm 0.21$ | $66.24 \pm 0.24$ |
| 5-Shot | $84.24 \pm 0.13$ | $84.52 \pm 0.16$ |

## 5 MAKING MAML PERMUTATION-INVARIANT DURING META-TESTING

We study approaches that can make MAML permutation-invariant during meta-testing. That is, we take the same learned initialization $\boldsymbol{\theta}$ as in section 4 without changing the meta-training phase.

We first investigate **searching for the best permutation for each task**. As we cannot access query sets' labels, we use the support sets' data as a proxy. We choose the best permutation according to which permutation, either before or after inner loop updates (less practical), leads to the highest accuracy or smallest loss on the support set. Table 1 summarizes the results: none of them leads to consistent gains. We hypothesize two reasons. First, due to mutually-exclusive tasks in meta-training, the learned $\boldsymbol{\theta}$ by MAML would produce chance-level predictions before updates (see Appendix C). Second, after updates, the support set accuracy quickly goes to $100\%$ and is thus not informative.

In stead of choosing one from many, we further explore **taking ensemble** (Breiman, 1996; Zhou, 2012; Dietterich, 2000) **over the predictions made by updated models of different permutations.** We note that this makes MAML permutation-invariant but inevitably needs more computations. To make the ensemble process clear, we permute the weight vectors in $\{\boldsymbol{w}_c\}_{c=1}^N$ rather than the class label assignments in a task: the two methods are equivalent but the former is easier for aggregating predictions. We study two variants: (a) full permutations (*i.e.*, 120 of them in five-way tasks), which is intractable for larger $N$; (b) rotated permutations, which rotates the index $c$ in $\{\boldsymbol{w}_c\}_{c=1}^N$[3], leading to $N$ permutations. Table 2 shows the results — ensemble consistently improves MAML. Even with the rotated version that has much fewer permutations than the full version, the gains are comparable.

*We emphasize that our focus here is to make MAML permutation-invariant in meta-testing, not to explore and compare all potential ways of performing ensemble on MAML.*

We further explore an efficient approach to make MAML permutation-invariant, which is to **manipulate the learned initialization of** $\{\boldsymbol{w}_c\}_{c=1}^C$. Concretely, MAML is sensitive to the permutations in class assignments because $\boldsymbol{w}_c \neq \boldsymbol{w}_{c'}$ for $c \neq c'$. One method to overcome this is to make $\boldsymbol{w}_c = \boldsymbol{w}_{c'}$ during meta-testing. Here, we investigate re-initializing each $\boldsymbol{w}_c$ by their average in meta-testing: $\boldsymbol{w}_c \leftarrow \frac{1}{N} \sum_{c'=1}^N \boldsymbol{w}_{c'}$. By doing so, no matter which permutation we perform, the model after inner loop optimization will be the same and lead to the same query set accuracy. Table 3 summarizes the results; this approach improves vanilla MAML (see Table 2) in three of the four cases.

At first glance, this approach may not make sense since the resulting $\{\boldsymbol{w}_c\}_{c=1}^C$, before inner loop updates, are identical and simply make random predictions. However, please note that even the original $\{\boldsymbol{w}_c\}_{c=1}^C$ have an averaged chance accuracy (cf. Figure 3). In Appendix E, we provide an explanation of this approach by drawing an analogy to dropout (Srivastava et al., 2014). In short, in meta-training, we receive a task with an arbitrary permutation, which can be seen as drawing a permutation at random for the task. In meta-testing, we then take expectation over permutations, which essentially lead to the averaged $\boldsymbol{w}_c$.

---

[3]That is, we consider re-assign $\boldsymbol{w}_c$ to $\boldsymbol{w}_{(c+\gamma \mod N)+1}$, where $\gamma \in [N]$.

Table 4: 5-Way 1-Shot and 5-Shot classification accuracy and 95% confidence interval on *Mini*ImageNet and *Tiered*ImageNet (over 10,000 tasks), using ResNet-12 as the backbone. †: MAML with 5 inner loop steps in meta-training/testing. ⋆: we carefully select the number of inner loop steps, based on the meta-validation set.

| Dataset → | *Mini*ImageNet | | *Tiered*ImageNet | |
|---|---|---|---|---|
| Setups → | 1-Shot | 5-Shot | 1-Shot | 5-Shot |
| ProtoNet (Snell et al., 2017) | $62.39 \pm 0.20$ | $80.53 \pm 0.20$ | $68.23 \pm 0.23$ | $84.03 \pm 0.16$ |
| ProtoMAML (Triantafillou et al., 2020) | $64.12 \pm 0.20$ | $81.24 \pm 0.20$ | $68.46 \pm 0.23$ | $84.67 \pm 0.16$ |
| MetaOptNet (Lee et al., 2019) | $62.64 \pm 0.35$ | $78.63 \pm 0.68$ | $65.99 \pm 0.72$ | $81.56 \pm 0.53$ |
| MTL+E3BM (Sun et al., 2019) | $63.80 \pm 0.40$ | $80.10 \pm 0.30$ | $71.20 \pm 0.40$ | $85.30 \pm 0.30$ |
| RFS-Distill (Tian et al., 2020) | $64.82 \pm 0.60$ | $82.14 \pm 0.43$ | $69.74 \pm 0.72$ | $84.41 \pm 0.55$ |
| DeepEMD (Zhang et al., 2020) | $65.91 \pm 0.82$ | $82.41 \pm 0.56$ | $\mathbf{71.52 \pm 0.69}$ | $86.03 \pm 0.49$ |
| MATE+MetaOpt (Chen et al., 2020) | $62.08 \pm 0.64$ | $78.64 \pm 0.46$ | $71.16 \pm 0.87$ | $86.03 \pm 0.58$ |
| DSN-MR (Simon et al., 2020) | $64.60 \pm 0.72$ | $79.51 \pm 0.50$ | $67.39 \pm 0.82$ | $82.85 \pm 0.56$ |
| FEAT (Ye et al., 2020a) | $\mathbf{66.78 \pm 0.20}$ | $82.05 \pm 0.14$ | $70.80 \pm 0.23$ | $84.79 \pm 0.16$ |
| MAML (5-Step†) | $62.90 \pm 0.20$ | $80.81 \pm 0.14$ | $59.08 \pm 0.20$ | $80.04 \pm 0.16$ |
| MAML (our reimplementation⋆) | $64.42 \pm 0.20$ | $83.44 \pm 0.14$ | $65.72 \pm 0.20$ | $84.37 \pm 0.16$ |
| UNICORN-MAML | $65.17 \pm 0.20$ | $\mathbf{84.30 \pm 0.14}$ | $69.24 \pm 0.20$ | $\mathbf{86.06 \pm 0.16}$ |

## 6 UNICORN-MAML: LEARNING A SINGLE WEIGHT VECTOR

The experimental results in section 5 are promising: by making MAML permutation-invariant in meta-testing, we can potentially improve vanilla MAML. While ensemble inevitably increases the computational burdens, the method by manipulating the learned initialization of $\{w_c\}_{c=1}^N$ keeps the same run time as vanilla MAML. In this section, we investigate the latter approach further. We ask:

> If we directly learn *a single weight vector* $w$ to initialize $\{w_c\}_{c=1}^N$ in meta-training, making the inner loop optimization in meta-training and meta-testing *consistent* and both *permutation-invariant*, can we further improve the accuracy?

Concretely, we redefine the learnable meta-parameters $\theta$ of MAML, which become $\theta = \{\phi, w\}$, where $\phi$ is for the feature extractor. We name this method **UNICORN-MAML**, as we meta-train only a single weight vector $w$ in the classification head. The inner loop optimization and outer loop optimization of UNICORN-MAML very much follow MAML, with some slight changes.

- **Inner loop optimization:** At the beginning, $w$ is duplicated into $\{w_c = w\}_{c=1}^N$. That is, we use $w$ to initialize every $w_c$, $\forall c \in [N]$. These $\{w_c = w\}_{c=1}^N$ then undergo the same inner loop optimization process as vanilla MAML (cf. Equation 1).
- **Outer loop optimization:** Let us denote the updated model by $\theta' = \{\phi', w'_1, \cdots, w'_N\}$. To perform the outer loop optimization for $w$ in meta-training, we collect the gradients derived from the query set $\mathcal{Q}$. Let us denote by $\nabla_{w_c} \mathcal{L}(\mathcal{Q}, \theta')$ the gradient w.r.t. the initial $w_c$ (cf. subsection 2.2). Since $w_c$ is duplicated from $w$, we obtain the gradient w.r.t. $w$ by $\sum_{c \in [N]} \nabla_{w_c} \mathcal{L}(\mathcal{Q}, \theta')$.

Table 4 summarizes the results of UNICORN-MAML, MAML, and many existing few-shot learning algorithms. UNICORN-MAML consistently improves MAML: on *Mini*ImageNet, UNICORN-MAML has a 0.7% gain on one-shot tasks and a 0.8% gain on five-shot tasks; on *Tiered*ImageNet, UNICORN-MAML has significant improvements (a 3.5% gain on one-shot tasks and a 1.6% gain on five-shot tasks). More importantly, UNICORN-MAML performs on a par with many recent algorithms on one-shot tasks, and achieves the highest accuracy on five-shot tasks. Specifically, compared to ProtoMAML and MetaOptNet, which are both permutation-invariant variants of MAML (see the related work paragraph at the end of this section), UNICORN-MAML notably outperforms them.

**Other results.** We evaluate UNICORN-MAML on **CUB** (Wah et al., 2011) and use the ConvNet backbone on *Mini*ImageNet in the appendix. UNICORN-MAML achieves promising improvements.

**Why does UNICORN-MAML work?** The design of UNICORN-MAML ensures that, without using the support set to update the model, the model simply performs at the chance level on the query set. In other words, its formulation inherently helps prevent memorization over-fitting (Yin et al., 2020).

**Embedding adaptation is needed.** We analyze UNICORN-MAML in terms of its inner loop updates during meta-testing, similar to Figure 3. This time, we also investigate updating or freezing the

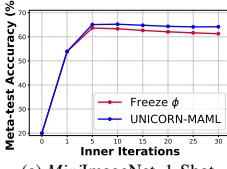 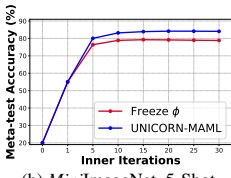 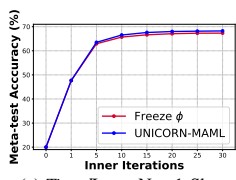 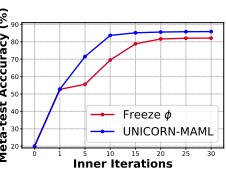

| (a) *Mini*ImageNet, 1-Shot | (b) *Mini*ImageNet, 5-Shot | (c) *Tiered*ImageNet, 1-Shot | (d) *Tiered*ImageNet, 5-Shot |

Figure 5: The change of meta-testing accuracy (over 10,000 tasks) along with the process of inner loop updates based on UNICORN-MAML. We investigate updating/freezing the feature extractor $\phi$.

Table 5: 5-Way 10/20/30/50-Shot classification accuracy on *Mini*ImageNet over 10,000 tasks with a ResNet backbone. (The confidence interval is omitted due to space limit.)

| Multi-Shot | 10-shot | 20-shot | 30-shot | 50-shot |
|---|---|---|---|---|
| SimpleShot (Wang et al., 2019) | 84.89 | 86.91 | 87.53 | 88.08 |
| ProtoNet (Snell et al., 2017) | 82.83 | 84.61 | 85.07 | 85.57 |
| FEAT (Ye et al., 2020a) | 85.15 | 87.09 | 87.82 | 87.83 |
| MAML (our implementation) | 88.08 | 90.23 | 91.06 | 92.14 |
| UNICORN-MAML | **88.38** | **90.96** | **91.96** | **92.86** |

Table 6: 5-Way 1/5-Shot classification accuracy on CUB based on the best learned model with a ResNet backbone from *Mini*ImageNet.

| *Mini*ImageNet → CUB | 1-Shot | 5-Shot |
|---|---|---|
| Baseline++ (Chen et al., 2019) | 50.37 | 73.30 |
| ProtoNet (Snell et al., 2017) | 50.01 | 72.02 |
| Neg-Cosine (Liu et al., 2020) | 47.74 | 69.30 |
| MAML (our implementation) | 51.25 | 73.86 |
| UNICORN-MAML | **51.80** | **75.67** |

feature extractor $\phi$. Figure 5 shows the results on five-way one- and five-shot tasks on both datasets. UNICORN-MAML's accuracy again begins with 20% but rapidly increases along with the inner loop updates. In three out of four cases, adapting the feature extractor $\phi$ is necessary for claiming a higher accuracy, even if the backbone has been well pre-trained, which aligns with the recent claim by Arnold & Sha (2021): *"Embedding adaptation is still needed for few-shot learning."*

**Experiments on larger shots and transferability.** MAML or similar algorithms that involve a bi-level optimization problem (*i.e.*, inner loop and outer loop optimization) are often considered more complicated and computationally expensive than algorithms without bi-level optimization, such as ProtoNet (Snell et al., 2017) or SimpleShot (Wang et al., 2019). Nevertheless, the inner loop optimization does strengthen MAML's adaptability during meta-testing, especially (a) when the meta-testing tasks are substantially different from the meta-training tasks (*e.g.*, meta-training using *Mini*ImageNet but meta-testing on CUB) or (b) when the number of shots increases (*e.g.*, from $1 \sim 5$ to $10 \sim 50$). In Table 5 and Table 6, we conduct further experiments to justify these aspects.

**Related work.** We note that some variants of MAML are permutation-invariant, even though they are not designed for the purpose. For example, LEO (Rusu et al., 2019) computes class prototypes (*i.e.*, averaged features per class) to encode each class and uses them to produce task-specific initialization. However, it introduces additional sub-networks. MetaOptNet (Lee et al., 2019) performs inner loop optimization only on $\{w_c\}_{c=1}^{N}$ (till convergence), making it a convex problem which is not sensitive to the initialization and hence the permutations. This method, however, has a high computational burden and needs careful hyper-parameter tuning for the additionally introduced regularizers. Proto-MAML (Triantafillou et al., 2020) initializes the linear classifiers $\{w_c\}_{c=1}^{N}$ with the prototypes, which could be permutation-invariant but cannot achieve accuracy as high as our UNICORN-MAML.

# 7 DISCUSSION AND CONCLUSION

There have been an abundance of "novel" algorithms proposed for few-shot classification (Hospedales et al., 2020; Wang et al., 2020). In these papers, MAML (Finn et al., 2017) is frequently considered as a baseline, but shows inferior results. This raises our interests. Is it because MAML is not suitable for few-shot classification, or is it because MAML has not been applied appropriately to the problem?

We thus conduct a series of analyses on MAML for few-shot classification, including hyper-parameter tuning and the sensitivity to the permutations of class label assignments in few-shot tasks. We find that by using a large number of inner loop gradient steps (in both meta-training and meta-testing), MAML can achieve comparable results to many existing algorithms. By further making MAML permutation-invariant to the class label assignments, we present UNICORN-MAML, which outperforms many existing algorithms on five-shot tasks, without the need to add extra sub-networks. We hope that UNICORN-MAML could serve as a strong baseline for future work in few-shot classification.

## ACKNOWLEDGMENTS

This research is supported by National Key R&D Program of China (2020AAA0109401), CCF-Baidu Open Fund (2021PP15002000), NSFC (61773198, 61921006, 62006112, 62176117), Collaborative Innovation Center of Novel Software Technology and Industrialization, NSF of Jiangsu Province (BK20200313), NSF IIS-2107077, NSF OAC-2118240, NSF OAC-2112606, and the OSU GI Development funds. We are thankful for the generous support of computational resources by Ohio Supercomputer Center and AWS Cloud Credits for Research. We thank Sébastien M.R. Arnold (USC) for helpful discussions.

## REPRODUCIBILITY STATEMENT

The details of datasets, model architectures, hyper-parameters, and evaluation metrics are described in subsection 2.3 and Appendix B. Our code is available at `https://github.com/Han-Jia/UNICORN-MAML`, including the initialization weights pre-trained on the meta-training set and the checkpoints.

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

# Appendix

We provide contents that we omit in the main text.

- Appendix A: Additional background and related work.
- Appendix B: details of experimental setups (cf. section 2).
- Appendix C: details and analyses of permutations of class label assignments (cf. section 4).
- Appendix D: Additional experimental results.
- Appendix E: Additional explanation of our studied methods.

## A    BACKGROUNDS ON META-LEARNING AND FEW-SHOT LEARNING

Training a model under data budgets is important in machine learning, computer vision, and many other application fields, since the costs of collecting data and labeling them are by no means negligible. This is especially the case for deep learning models in visual recognition (He et al., 2016; Dosovitskiy et al., 2021; Simonyan & Zisserman, 2015; Szegedy et al., 2015; Krizhevsky et al., 2012; Huang et al., 2017), which usually need thousands of, millions of, or even more images to train (Russakovsky et al., 2015; Deng et al., 2009; Guo et al., 2016; Thomee et al., 2016; Mahajan et al., 2018; Joulin et al., 2016) in a conventional supervised manner. Different from training a model to predict at the *instance* level, meta-learning attempts to learn the inductive bias across training *tasks* (Baxter, 2000; Vilalta & Drissi, 2002). More specifically, meta-learning aims to train a "meta-model" to summarize the common characteristics of tasks and generalize them to those *novel* but related tasks (Maurer, 2009; Maurer et al., 2016; Denevi et al., 2018; Ye et al., 2021). Meta-learning has been applied in various fields, including few-shot learning (Ravi & Larochelle, 2017; Snell et al., 2017; Wang & Hebert, 2016; Ye et al., 2020a;b; Sung et al., 2018; Zhang et al., 2018a; Wang et al., 2018b; Tseng et al., 2020; Fei et al., 2021), optimization (Andrychowicz et al., 2016; Wichrowska et al., 2017; Li & Malik, 2017; Bello et al., 2017), reinforcement and imitation learning (Stadie et al., 2018; Frans et al., 2018; Wang et al., 2017a; Duan et al., 2016; 2017; Yu et al., 2018), unsupervised learning (Garg & Kalai, 2018; Metz et al., 2019; Edwards & Storkey, 2017; Reed et al., 2018), continual learning (Riemer et al., 2019; Kaiser et al., 2017; Al-Shedivat et al., 2018), imbalance learning (Wang et al., 2017c; Ren et al., 2018b), transfer and multi-task learning (Motiian et al., 2017; Balaji et al., 2018; Ying et al., 2018; Zhang et al., 2018b; Li et al., 2019; 2018), active learning (Ravi & Larochelle, 2018; Sharma et al., 2018; Bachman et al., 2017; Pang et al., 2018), data compression (Wang et al., 2018a), architecture search (Elsken et al., 2019), recommendation systems (Vartak et al., 2017), data augmentation (Ratner et al., 2017), teaching (Fan et al., 2018), hyper-parameter tuning (Franceschi et al., 2017; Probst et al., 2019), etc.

In few-shot learning (FSL), meta-learning is applied to learn the ability of *"how to build a classifier using limited data"* that can be generalized across tasks. Such an inductive bias is first learned over few-shot tasks composed of "base" classes, and then evaluated on tasks composed of "novel" classes. For example, few-shot classification can be implemented in a non-parametric way with soft nearest neighbor (Vinyals et al., 2016) or nearest center classifiers (Snell et al., 2017), so that the feature extractor is learned and acts at the task level. The learned features pull similar instances together and push dissimilar ones far away, such that a test instance can be classified even with a few labeled training examples (Koch et al., 2015). Considering the complexity of a hypothesis class, the model training configurations (*i.e.*, hyper-parameters) also serve as a type of inductive biases. Andrychowicz et al. (2016); Ravi & Larochelle (2017) meta-learn the optimization strategy for each task, including the learning rate and update directions. Other kinds of inductive biases are also explored. Hariharan & Girshick (2017); Wang et al. (2018b) learn a data generation prior to augment examples given few images; Dai et al. (2017) extract logical derivations from related tasks; Wang et al. (2017b); Shyam et al. (2017) learn the prior to attend images.

Model-agnostic meta-learning (MAML) (Finn et al., 2017) proposes another inductive bias, *i.e.*, the model initialization. After the model initialization that is shared among tasks has been meta-trained, the classifier of a new few-shot task can be fine-tuned with several steps of gradient descent from that initial point. The universality of this MAML-type updates is proved in (Finn & Levine, 2018). MAML has been applied in various scenarios, such as uncertainty estimation (Finn et al., 2018),

robotics control (Yu et al., 2018; Clavera et al., 2019), neural translation (Gu et al., 2018), language generation (Huang et al., 2018), etc.

Despite the success, there are still problems with MAML. For example, Nichol et al. (2018) handle the computational burden by presenting a family of approaches using first-order approximations. Rajeswaran et al. (2019) propose to leverage implicit differentiation, making the calculation of the meta-gradients much efficient and accurate. Antoniou et al. (2019) provide a bunch of tricks to train and stabilize the MAML framework. Bernacchia (2021) points out that negative rates of gradient updates help in some scenarios. Rajendran et al. (2020); Yao et al. (2021); Yin et al. (2020) argue that the learned initialization by MAML may be at high risk of (a) memorization over-fitting, where it solves meta-training tasks without the need of inner loop optimization, or (b) learner over-fitting, where it over-fits to the meta-training tasks and fails to generalize to the meta-testing tasks. They thus propose to improve MAML by imposing a regularizer or performing data augmentation.

Since MAML applies a uniform initialization to all the tasks (*i.e.*, the same set of $\{w_c\}_{c=1}^N$ and $\phi$), recent methods explore ways to better incorporate task characteristics. Lee et al. (2019); Bertinetto et al. (2019); Raghu et al. (2020) optimize the linear classifiers $\{w_c\}_{c=1}^N$ (not the feature $f_\phi$) till convergence in the inner loop; Triantafillou et al. (2020); Ye et al. (2020b) initialize the linear classifiers using class prototypes (*i.e.*, aggregated features per class) so they are task-aware even before the inner loop optimization. Another direction is to enable task-specific initialization for the entire model (Requeima et al., 2019; Vuorio et al., 2019; Yao et al., 2019; Rusu et al., 2019), which often needs additional sub-networks.

Our work is complementary to the above improvements of MAML: we find an inherent permutation issue of MAML in meta-testing and conduct a detailed analysis. We then build upon it to improve MAML. We note that some of the above methods can be permutation-invariant even though they are not designed for the purpose. For example, LEO (Rusu et al., 2019) computes class prototypes (*i.e.*, averaged features per class) to represent each semantic class[4]. However, it introduces additional sub-networks. MetaOptNet (Lee et al., 2019) performs inner loop optimization only on $\{w_c\}_{c=1}^N$ (till convergence), making it a convex problem which is not sensitive to the initialization and hence the permutations. This method, however, has a high computational burden and needs careful hyper-parameter tuning for the additionally introduced regularizers. Proto-MAML (Triantafillou et al., 2020) initializes the linear classifiers $\{w_c\}_{c=1}^N$ with the prototypes, which could be permutation-invariant but cannot achieve accuracy as high as our UNICORN-MAML. We note that one fundamental difference between our work and (Dhillon et al., 2020), LEO (Rusu et al., 2019), and Proto-MAML (Triantafillou et al., 2020) is the motivation: they aim to provide the initialization of the classifier head with better semantic meanings, while our goal is to resolve the sensitivity of MAML to the permutations.

## B    DETAILS OF EXPERIMENTAL SETUPS

We follow the pre-training procedure in (Ye et al., 2020a). During pre-training we append the feature extractor backbone with a fully-connected layer for classification, and train it to classify all classes in the base class set (*e.g.*, 64 classes in the *Mini*ImageNet) with the cross-entropy loss. In this stage, we apply standard ImageNet image augmentations (*e.g.*, random crop and random flip). The best pre-trained feature extractor (*i.e.*, epoch) is selected based on the one-shot classification accuracy on the meta-validation set. Specifically, we sample one-shot tasks from the meta-validation set, and apply the nearest neighbor classifier on top of the extracted features to evaluate the quality of the backbone. Finally, the best pre-trained backbone is used to initialize the feature extractor for MAML.

## C    PERMUTATIONS OF CLASS LABEL ASSIGNMENTS

We provide more analyses and discussions on the permutation issue in the class label assignment. As illustrated in Figure 1 (a), few-shot tasks of the same set of $N$ semantic classes (*e.g.*, "unicorn", "bee", etc.) can be associated with different label assignments (*i.e.*, $c \in [N]$) and are paired with the learned initialization $\{w_c\}_{c=1}^N$ of MAML differently. For five-way tasks, there are 120 permutations.

---

[4]We note that while Requeima et al. (2019); Vuorio et al. (2019); Yao et al. (2019) also enable task-specific initialization with additional sub-networks for task embedding, their methods cannot resolve the permutation issue. This is because they take an average of the feature embeddings over $N$ classes to represent a task.

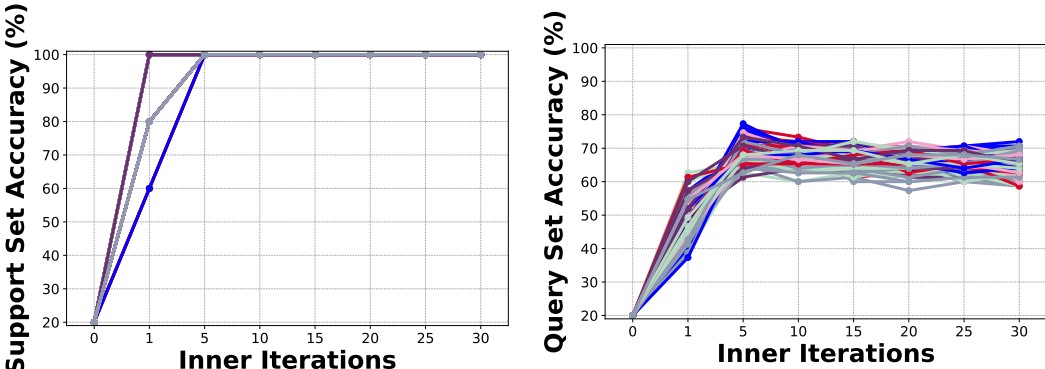

Figure F: The support (left) and query (right) set accuracy on a randomly sampled five-way one-shot meta-testing task from *Mini*ImageNet. We plot the accuracy of each permutation (totally 120) along with the process of inner loop optimization (the same permutation is colored the same in the left and right images).

In section 4, we study how the permutations affect the meta-testing accuracy (after the inner loop optimization of $\phi$ and $\{w_c\}_{c=1}^N$) and see a high variance among the permutations. We note that, the inner loop optimization updates not only the linear classifiers $\{w_c\}_{c=1}^N$, but also $\phi$ of the feature extractor. Different permutations therefore can lead to different feature extractors.

Here, we further sample a five-way one-shot meta-testing task, and study the change of accuracy along with the inner loop updates (using a MAML trained with a fixed number of inner loop updates). Specifically, we plot both the support set and query set accuracy for each permutation. As shown in Figure F, there exists a high variance of query set accuracy among permutations after the inner loop optimization. This is, however, not the case for the support set. (The reason that only three curves appear for the support set is because there are only five examples, and all the permutations reach 100% support set accuracy within five inner loop steps.) Interestingly, for all the permutations, their initialized accuracy (*i.e.*, before inner loop optimization) is all 20%. After an investigation, we find that the meta-learned $\{w_c\}_{c=1}^N$ (initialization) is dominated by one of them; *i.e.*, all the support or query examples are classified into one class. While this may not always be the case for other few-shot tasks or if we re-train MAML, for the task we sampled, it explains why we obtain 20% for all permutations. We note that, even with an initial accuracy of 20%, the learned initialization can be updated to attain high classification accuracy.

We further compare the change of support and query set accuracy along with the inner loop optimization in Figure F. We find that, while both accuracy increases, since the support set accuracy converges quickly and has a smaller variance among permutations, it is difficult to use its information to determine which permutation leads to the highest query set accuracy. This makes sense since the support set is few-shot: its accuracy thus cannot robustly reflect the query set accuracy. This explains why the methods studied in Table 1 cannot determine the best permutation for the query set.

### C.1 MATHEMATICAL EXPLANATION

We provide a simple mathematical explanation for why, on average, the query set accuracy is at the chance level if we directly apply the learned initialized model. Suppose we have a five-way task with five semantic classes {"dog", "cat", "bird", "car", "person"}. Without loss of generality, let us assume that the query set has only five examples, one from each class. Let us also assume that the best permutation — *i.e.*, the best assignment of $\{w_1, \cdots, w_5\}$ to these classes — gives a 100% query set accuracy using the initialized model. Since there are in total 120 possible permutations, there will be 10 of them with 60% accuracy (*i.e.*, by switching two-class indices), 20 of them with 40% accuracy (*i.e.*, by shuffling the indices of three classes such that they do not take their original indices), 45 of them with 20% accuracy, and 54 of them with 0% accuracy. Taking an average over these permutations gives a 20% accuracy. In other words, even if one of the permutations performs well, on average the accuracy will be close to random.

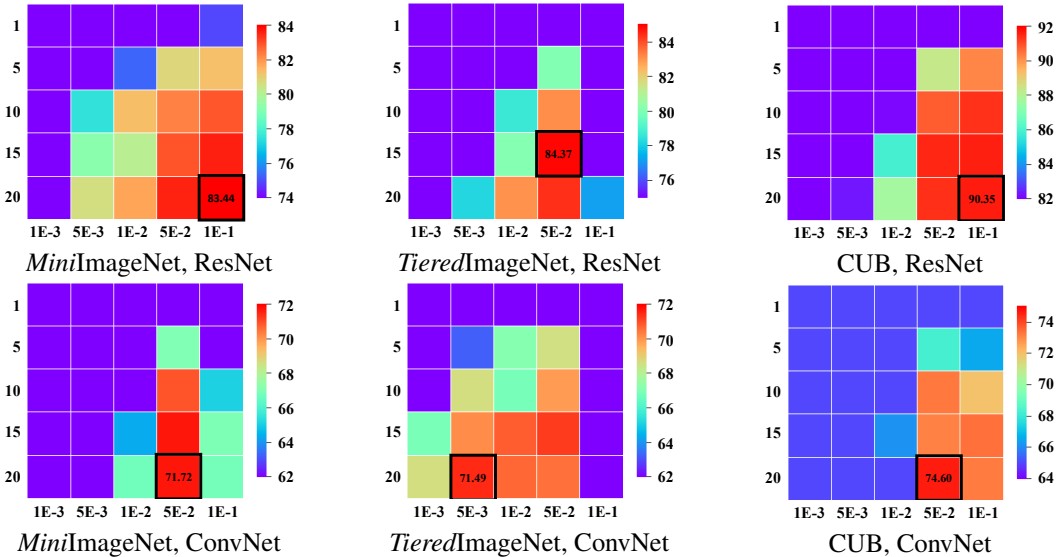

Figure G: Heat maps of MAML's five-way five-shot accuracy on *Mini*ImageNet, *Tiered*ImageNet, and CUB w.r.t. the inner loop learning rate $\alpha$ (x-axis) and the number of inner loop updates $M$ (y-axis). For each map, **we set accuracy below a threshold to a fixed value for clarity**; we denote the best accuracy by a black box.

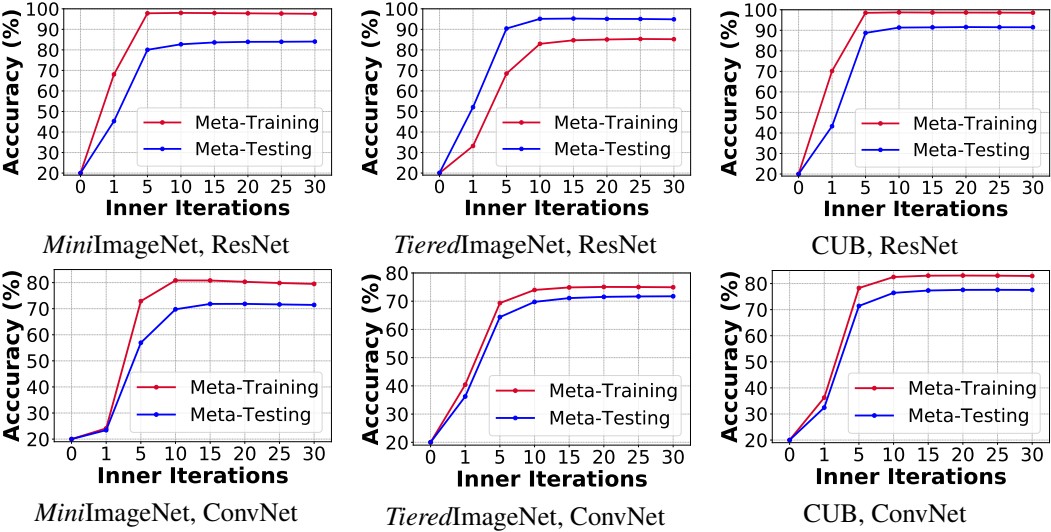

Figure H: We plot the change of the five-way five-shot classification accuracy (on the query set), averaged over 10,000 tasks sampled from either the meta-training (red) or meta-testing classes (blue), along with the process of inner loop updates, using the best model initialization learned by MAML.

# D  ADDITIONAL EXPERIMENTAL RESULTS

Similar to Figure 2, we plot the meta-testing accuracy of five-way five-shot tasks on the three datasets over ResNet and ConvNet backbones in Figure 2. We get a similar trend with Figure 2 where MAML achieves higher and much more stable accuracy (w.r.t. the learning rate) when $M$ is larger than $15$. Specifically, for *Mini*ImageNet with ResNet, the highest accuracy $83.44\%$ is obtained with $M = 20$, higher than $80.81\%$ with $M = 5$.

We also plot the change of five-way five-shot classification accuracy (on the query set), averaged over 10,000 tasks sampled from either the meta-training (red) or meta-testing classes (blue), along with the process of inner loop updates, using the best model initialization learned by MAML in Figure H for

Table G: 5-Way 1/5-Shot classification accuracy and 95% confidence interval on CUB, evaluated over 10,000 tasks with a ResNet-12 backbone. ‡: methods with a ResNet-18 backbone. †: We train MAML with 5 inner loop steps in both meta-training and meta-testing. ⋆: we carefully select the number of inner loop steps for MAML, based on the meta-validation set.

| ResNet-12 | 1-Shot | 5-Shot |
|---|---|---|
| MatchNet (Vinyals et al., 2016) | $66.09 \pm 0.92$ | $82.50 \pm 0.58$ |
| ProtoNet (Snell et al., 2017) | $71.87 \pm 0.85$ | $85.08 \pm 0.57$ |
| DeepEMD (Zhang et al., 2020) | $75.65 \pm 0.83$ | $88.69 \pm 0.50$ |
| Baseline++ (Chen et al., 2019)‡ | $67.02 \pm 0.90$ | $83.58 \pm 0.54$ |
| AFHN† (Li et al., 2020) | $70.53 \pm 1.01$ | $83.95 \pm 0.63$ |
| Neg-Cosine (Liu et al., 2020)‡ | $72.66 \pm 0.85$ | $89.40 \pm 0.43$ |
| Align (Afrasiyabi et al., 2020)‡ | $74.22 \pm 1.09$ | $88.65 \pm 0.55$ |
| MAML (5-Step†) | $76.53 \pm 0.20$ | $88.34 \pm 0.16$ |
| MAML (our reimplementation⋆) | $77.67 \pm 0.20$ | $90.35 \pm 0.16$ |
| UNICORN-MAML | $\mathbf{78.07 \pm 0.20}$ | $\mathbf{91.67 \pm 0.16}$ |

Table H: 5-Way 1-Shot and 5-Shot classification accuracy and 95% confidence interval on *Mini*ImageNet over 10,000 tasks with a four-layer ConvNet backbone. †: We train MAML with 5 inner loop steps in both meta-training and meta-testing. ⋆: we carefully select the number of inner loop steps for MAML, based on the meta-validation set.

| ConvNet | 1-Shot | 5-Shot |
|---|---|---|
| MAML (Finn et al., 2017) | $48.70 \pm 1.84$ | $63.11 \pm 0.92$ |
| MAML++ (Antoniou et al., 2019) | $52.15 \pm 0.26$ | $68.32 \pm 0.44$ |
| Reptile (Nichol et al., 2018) | $49.97 \pm 0.32$ | $65.99 \pm 0.58$ |
| FEAT (Ye et al., 2020a) | $55.15 \pm 0.20$ | $71.61 \pm 0.16$ |
| KTN (visual) (Peng et al.) | $54.61 \pm 0.80$ | $71.21 \pm 0.66$ |
| PARN (Wu et al., 2019) | $55.22 \pm 0.84$ | $71.55 \pm 0.66$ |
| MAML-MMCF (Yao et al., 2021) | $50.35 \pm 1.82$ | $64.91 \pm 0.96$ |
| MAML (5-Step†) | $53.15 \pm 0.20$ | $67.01 \pm 0.16$ |
| MAML (our reimplementation⋆) | $54.89 \pm 0.20$ | $71.72 \pm 0.16$ |
| UNICORN-MAML | $\mathbf{55.70 \pm 0.20}$ | $\mathbf{72.68 \pm 0.16}$ |

each pair of dataset and backbone. There indicates a similar phenomenon as Figure 3, where MAML needs a large $M$.

We further evaluate UNICORN-MAML on CUB dataset with ResNet-12 backbone and on *Mini*ImageNet with the four-layer ConvNet backbone. The results are listed in Table G and Table H, respectively. By comparing UNICORN-MAML with others, we find the carefully tuned MAML shows promising results and UNICORN-MAML outperforms the existing methods.

# E  ADDITIONAL EXPLANATIONS OF OUR STUDIED METHODS

We provide some more explanations on the ensemble and forced permutation-invariant methods introduced in section 5. For the ensemble method, given a few-shot task, we can permute $\{w_c\}_{c=1}^N$ to pair them differently with such a task. We can then perform different inner loop optimizations to obtain a set of five-way classifiers that we can perform ensemble upon. In the main text, we average the posterior probabilities of these five-way classifiers to make the final predictions.

Since the permutation affects the meta-training phase as well, we can interpret the meta-training phase as follows. Ever time we sample a few-shot task $\mathcal{T} = (\mathcal{S}, \mathcal{Q})$, we also sample a permutation $\pi : [N] \mapsto [N]$ to re-label the classes. (We note that, this is implicitly done when few-shot tasks are sampled.) We then take $\mathcal{T}_\pi = (\mathcal{S}_\pi, \mathcal{Q}_\pi)$ to optimize $\boldsymbol{\theta}$ in the inner loop. That is, in meta-training, the

objective function in Equation 2 can indeed be re-written as

$$\mathbb{E}_{(\mathcal{S},\mathcal{Q})\sim p(\mathcal{T}),\pi\sim p(\pi)} \mathcal{L}(\mathcal{Q}_\pi, \boldsymbol{\theta}') = \mathbb{E}_{(\mathcal{S},\mathcal{Q})\sim p(\mathcal{T}),\pi\sim p(\pi)} \mathcal{L}(\mathcal{Q}_\pi, \mathsf{InLoop}(\mathcal{S}_\pi, \boldsymbol{\theta}, M)), \qquad \text{(D)}$$

where $p(\pi)$ is a uniform distribution over all possible permutations. Equation D can be equivalently re-written as

$$\mathbb{E}_{(\mathcal{S},\mathcal{Q})\sim p(\mathcal{T}),\pi\sim p(\pi)} \mathcal{L}(\mathcal{Q}, \boldsymbol{\theta}'_\pi) = \mathbb{E}_{(\mathcal{S},\mathcal{Q})\sim p(\mathcal{T}),\pi\sim p(\pi)} \mathcal{L}(\mathcal{Q}, \mathsf{InLoop}(\mathcal{S}, \boldsymbol{\theta}_\pi, M)), \qquad \text{(E)}$$

where $\boldsymbol{\theta}_\pi$ means that the initialization of the linear classifiers $\{\boldsymbol{w}_c\}_{c=1}^N$ are permuted; $\boldsymbol{\theta}'_\pi$ is the corresponding updated model. This additional *sampling process* of $\pi$ is reminiscent of dropout (Srivastava et al., 2014), which randomly masks out a neural network's neurons or edges to prevent an over-parameterized neural network from over-fitting. During testing, dropout takes expectation over the masks. We also investigate a similar idea during meta-testing, by taking expectation (*i.e.*, average) on the learned initiation of the linear classifiers over different permutations. This results in a new initialization during the meta-testing phase: $\boldsymbol{w}_c \leftarrow \frac{1}{N} \sum_{c'=1}^N \boldsymbol{w}_{c'}$.

