# OpenReview forum: "How to Train Your MAML to Excel in Few-Shot Classification"
_ICLR.cc/2022/Conference — ICLR 2022 Poster_

### Official Review · Reviewer_zwrd · 2021-10-29

**Correctness:** 4
**Technical Novelty And Significance:** 3
**Empirical Novelty And Significance:** 4
**Recommendation:** 8
**Confidence:** 4

**Main Review:**

### Strengths

- The paper is well written and easy to follow. The main problem at hand, the random label assignment during evaluation, is introduced through different experimentations, showing why and where is the problem in MAML.
- A good explanation and novel investigation of this problem. The results show that MAML can perform on-par with and even outperforms other state-of-the-art meta-learning methods.
- The experiments are detailed and multiple methods are tested to try to alleviate the permutation label assignment problem for MAML, before introducing their proposed modification to the algorithm.
- The experimental settings and hyperparameters are all mentioned and documented to help reproducibility.
- Even though the experiments focuses on the MAML algorithm, the authors compare their results with related works, but also discuss how other methods deal with the random label assignment.

### Weaknesses

- I am quite surprised by the performance they achieve with their vanilla MAML, even without considering the increased inner steps. I never saw performance as high in other papers using MAML with a ResNet12. Maybe the difference come from the pretrained weights ? I didn't find any information on how the backbone was pretrained though and I think this information would help for reproducibility.

**Summary Of The Paper:**

The paper mainly investigates the effect of permutation in the class label assignment in the tasks for the MAML algorithm. First, the authors show that MAML requires a higher number of inner loop updates than what is commonly used. Then, they show that MAML is sensible to the permutation of the class labels in the tasks and experimented with diverse methods to alleviate this problem. Finally, they proposed Unicorn-MAML, a modification of the MAML algorithm that learns a single weight vector for the classifier layer, to make the model permutation invariant. They show through experiments on two different datasets (mini-ImageNet and tiered-ImageNet) that it achieves or outperforms state-of-the-art performance.

**Summary Of The Review:**

I found the paper clear and easy to read. All the experiments are detailed as well as the thought process behind them. I would appreciate more information on the full training process to improve reproducibility.

---

> ### Author Response · Authors · 2021-11-21
> **Thank you for your detailed, positive, and encouraging review**
>
> We thank the reviewer for the positive and encouraging review. We provide our responses as follows.
>
> **The performance of the vanilla MAML.** You are correct: pre-training plays an important role in having a better vanilla MAML. Indeed, pre-training the backbone has become a standard practice in few-shot classification, but, surprisingly, many reported results on MAML did not use a pre-trained backbone. We thus decided to use a pre-trained backbone for a fair comparison to state-of-the-art few-shot classification algorithms (as mentioned in section 2.3). More specifically, we follow the pre-training procedure in (Ye et al., 2020a). We will release our pre-trained models and code upon acceptance.
>
> In detail, during pre-training we append the feature extractor backbone with a fully-connected layer for classification, and train it to classify all classes in the base class set (e.g., 64 classes in the MiniImageNet) with the cross-entropy loss. In this stage, we apply standard ImageNet image augmentations (e.g., random crop and random flip). The best pre-trained feature extractor (i.e., epochs) is selected based on the one-shot classification accuracy on the meta-validation set. Specifically, we sample one-shot tasks from the meta-validation set, and apply the nearest neighbor classifier on top of the extracted features to evaluate the quality of the backbone. Finally, the best pre-trained backbone is used to initialize the feature extractor for MAML.

---

### Official Review · Reviewer_Qepb · 2021-11-02

**Correctness:** 3
**Technical Novelty And Significance:** 3
**Empirical Novelty And Significance:** 3
**Recommendation:** 8
**Confidence:** 4

**Main Review:**

Advantages

1. The paper is interesting for me. I believe it is worth questioning the key settings in the benchmark or classical baseline methods. This encourages people to rethink why the model works or not in specific scenarios. Also, the idea of UNICORN-MAML is simple and easy to implement.

2. This paper offers a detailed exploration for two factors (in MAML based meta-learning benchmark) that are ignored by previous studies: one is the number of inner-updates in each training/testing task and the other is the task label assignment in each task. It is interesting to see the statistics in Fig4 that permutations make such a big difference to specific test tasks.

Disadvantages

1. From the results in Table 2,5,6, it is pity that there is little concrete performance boost by using the new MAML (UNICORN-MAML) compared to MAML (ours) --- where I assume ours means "our careful implementation of MAML".

2. Main conclusions or results are from arbitrarily designed experiments on existing benchmarks of few-shot learning---small tasks of object classification. It is hard to guarantee that the conclusion maintains in meta-learning larger-scale tasks or in another new benchmark.

3. MAML was implemented originally not only on few-shot classification tasks but also on reinforcement learning tasks. While it is clear that this new UNICORN-MAML may not be straightforwardly used for RL, as it studies the specific problems in the image classification settings.

**Summary Of The Paper:**

This paper is an interesting "re"search on a classical meta-learning method MAML. It shows that both the inner update iterations and task label assignment have a clear influence on the performance of MAML. It raises some questions and uses a set of in-depth experiments to draw conclusions.

**Summary Of The Review:**

I think it is interesting to research and rethink the possible issues in the existing baselines or benchmarks. I have a few concerns about the significance of this current version of the paper because of the poor performance or the arbitrary design of experiments. I would like to see the comments of other reviewers and the discussion with the authors to make a final decision.

---

> ### Author Response · Authors · 2021-11-21
> **Thank you for your detailed and positive review**
>
> We thank the reviewer for the detailed and positive review. We provide our responses as follows.
>
> **1. Little concrete performance boost.** Thank you for the question. You are correct: “ours” in Tables 5 and 6 means “our careful implementation of MAML”. While the gaps are not always large in Tables 4, 5, and 6, UNICORN-MAML does consistently outperform MAML (our careful implementation of MAML), with a very simple but effective modification without introducing extra learnable parameters. It is also worth mentioning that MAML (our careful implementation of MAML) significantly outperforms MAML (5-Step+) in Table 4, which is based on our first contribution.
>
> **2. Arbitrarily designed experiments on existing benchmarks of few-shot learning.** In most parts of the paper, we strictly follow the experimental setups of existing few-shot classification benchmarks; Table 6 is also a standard task according to (Chen et al., 2019, Liu et al., 2020). We experimented with three benchmark datasets, miniImageNet, tiredImageNet, and CUB, which are considered much larger-scale and more challenging than Omniglot and CIFAR. Compared to the literature, we think the use of these three datasets is considered sufficient for few-shot classification, but we will be happy to explore other datasets. We also conduct detailed analysis and ablation studies of our approaches, and we respectfully think that they are not arbitrarily designed.
>
> **3. Limited to few-shot classification.** Thank you for the question. We totally agree that MAML is versatile and has achieved promising results in many applications, as mentioned in the introduction and appendix. We specifically focus on few-shot classification because it is one of the most popular tasks in meta-learning, but MAML has so far fallen far behind the state of the art. We hope our focused study would not downgrade our contribution.

---

### Official Review · Reviewer_mgm6 · 2021-11-03

**Correctness:** 4
**Technical Novelty And Significance:** 4
**Empirical Novelty And Significance:** 4
**Recommendation:** 8
**Confidence:** 5

**Main Review:**


I really liked the way that the authors study the MAML algorithm. They performed very systematic and reasonable experiments. The second part of the paper that looks at the permutation of the labels is interesting. Ideally, this should not have happened since the task is the same and labels should not impact the performance. For example, for metric-based approaches such as ProtoNets[2], this never happens. Also, the proposed approach (UNICORN-MAML) just defines a weight vector that is shared for all classes such that permutation has no impact on the updates. It is a very interesting way to solve the problem.








[1] Hsu K, Levine S, Finn C. Unsupervised Learning via Meta-Learning. In Int'l Conf. on Learning Representations 2018.

[2] Snell J, Swersky K, Zemel R. Prototypical networks for few-shot learning. In Proc. of the 31st Int'l Conf. on Neural Information Processing Systems 2017.


**Summary Of The Paper:**

The authors study the MAML algorithm and propose two ways to improve the performance. First, they study the number of inner gradient steps. Second, they look at the permutation of labels when learning a new task which ideally should not make any difference. However, it seems that during testing this could lead to different test performances on a target task. Based on this observation, they propose a novel and very interesting solution to share the weights of the classification layer (which they call UNICORN MAML).

**Summary Of The Review:**

Finally, I think this is a thorough study of a very useful algorithm and can improve many papers that are based on this approach. It is a good improvement on MAML that is based on a very detailed systematic study. As a result, I vote for acceptance.

---

> ### Author Response · Authors · 2021-11-21
> **Thank you for your detailed, positive, and encouraging review**
>
> We thank the reviewer for the detailed, positive, and encouraging review. We are pleased that the reviewer voted our paper for acceptance.

---

### Official Review · Reviewer_eimD · 2021-11-04

**Correctness:** 3
**Technical Novelty And Significance:** 2
**Empirical Novelty And Significance:** 2
**Recommendation:** 3
**Confidence:** 5

**Main Review:**

This is an interesting study which is mainly inspired by the empirical results. However, I have some concerns that needs to be addressed:




- In equation (1), the gradient is taken with respect to the $\theta^{‘}$ as we are in the inner-loop?




- One main concern of this study is that when MAML pairs the learned classifier head with different permutations of a task, there could be an inconsistency in the performance. However, I believe this limitation does not exist in works like ProtoNet, as we have no $\omega$ there for pairing, and the prototype of each class does not depend on the class index but support set samples of that class. One major question is that, since ProtoNet has implicitly addressed this concern before, what is the *importance* of this study?




- Considering good pairing between meta-test task and classifier head is not a new idea in few-shot learning. For example, in Dhillon et. al. [1], they implicitly consider this by initializing the classifier’s head using the normalized logits for each class, inspired by cosine distance. However, this work uses the simple idea of repeating a meta-learned vector as the classifier's weights. I think you need to add this work and similar works to related work, and emphasize on the differences.




- My *major concern* is about your motivation for this study. Let’s have a quick look at the possibility of the permutation you discussed as motivation and illustrated in Figure 1.

Considering the meta-test set of miniImageNet, there are 20 classes on this set. Then for 5-way classification as we need to sample 5 classes to construct a task (or episode), the number of permutations of 20 classes taken 5 at a time is 20!/(20-5)! = 1860480. Among these, 5!=120 could be the task that has the same classes. So, you are planning for an event with a chance of 120/1860480=0.000064 which is really negligible. This gets even worse for other datasets like tieredImageNet and CUB as they have larger numbers of classes in meta-test, or even for real scenarios with larger unseen classes.

However, looking at your results, I think your solution is more general  and may be also considered as differences in the class attributes. So, you need to change your motivating example and make it more general. Although your solution answers, I think you have approached it with the wrong motivation.




- The procedure to produce the results in figure 3 is not clear to me. Do you use meta-validation performance to select M? Or do you just test different values of M for training and testing? Please elaborate on this.




- In section 5, when you propose simple solutions to make MAML permutation-invariant during meta-testing, what is the point of searching for the best permutation for a meta-training task? How is it going to affect the training of more powerful features and improving generalization?




- Based on the results in Table 4, UNICORN-MAML (which has a lower number of parameters w.r.t MAML) improves its performance. However, I am concerned when authors mention that they have compared with state-of-the-art, because they have not included recent works in the field, like [2], which shows produced results fall behind current state-of-the-art. I think it is more reasonable to mention that UNICORN-MAML can be considered as a strong baseline for future works, and I am ok with this.




- In section 6, I didn’t find the answer for the question “Why does UNICORN-MAML work?” informative enough. You may give more detail which backed up with results, or some facts from previous works, or simply omit this.




- The explanation of Figure 5 is not relevant to the main purpose of the paper.  As one of key observations in your work, you have mentioned that MAML needs lots of inner-loop updates in Figure 3. However, when providing similar results (as you mentioned in the main text) for UNICORN-MAML, you conclude that it aligns with previous work that “we need to also adapt the feature extractor”? How do you relate this finding to your intention in this work?

It needs to be related to the main context. Please elaborate on this.




- Table 6 does not provide a fair comparison with current algorithms for cross-domain few-shot classification. As mentioned in [3], in the same scenario (miniImageNet $\rightarrow$ CUB), a metric-based algorithm like RelationNet, achieves 57.77% in 1-shot. Using FT proposed in [3], it can also achieve up to 59.94%. However, you have reported 51.80% as SOTA result in this case?




References:

[1] A Baseline for Few-Shot Image Classification, ICLR 2020.

[2] MELR: Meta-Learning via Modeling Episode-Level Relationships for Few-shot Learning, ICLR 2021.

[3] Cross-Domain Few-shot Classification via Learned Features-wise Transformation, ICLR 2020.




**Summary Of The Paper:**

This paper analyses the well-known and well-studied MAML algorithm and raises two key observations. First one is requiring a high number of inner-loop updates, and the second one is the variation in the meta-test accuracy when permuting the indices of the classes. Then it proposes to simply meta-train a single vector and duplicate it as initialization for the classifier head to make MAML permutation-invariant and improve its generalization performance on meta-test tasks.

**Summary Of The Review:**

This work proposes a simple yet interesting algorithm which meta-trains just a single vector and duplicates it as initialization for the all classification head weights for various classes in MAML. It improves the performance of the MAML, however I have some serious concerns regarding the importance, the validity of the motivation and the fairness of the results for this study.

---

> ### Author Response · Authors · 2021-11-21
> **Thank you for the detailed review (Part 2)**
>
> **The explanation of Figure 5.** We thank the reviewer’s question. We include Figure 5 for a further understanding of UNICORN-MAML, and we will specify this in the final version. We note that the inner-loop optimization updates both the feature extractor and the classification head. Since UNICORN-MAML begins with a classification head where all the classes use the same weight vector, we think it is crucial to investigate if UNICORN-MAML needs more inner-loop steps solely to update the classification head or not. We will modify our texts to make the purpose of this study clearer.
>
> **Table 6 for cross-domain few-shot classification.** We thank the reviewer’s comment and reference. We will cite [3] in our final version. We found that 57.77% and 59.46% by RelationNet are on “5”-shot tasks not “1”-shot tasks, and our “5”-shot result is 75.67%. Nevertheless, we include Table 6 mainly to support the strength of MAML-like algorithms, not to claim the state of the art on this task.

---

> ### Author Response · Authors · 2021-11-21
> **Thank you for the detailed review (Part 1)**
>
> We thank the reviewer for the comments. In the following, we try our best to address all the reviewer’s concerns.
>
> **Equation (1).** Thanks for the comment. Yes, the gradient is with respect to θ’ and we will correct it.
>
> **The importance of this study.** Thank you for the comment. The contributions of this study, as summarized in the conclusion, are 1) a systematic understanding of why MAML, one of the most popular and general-purpose meta-learning algorithms, performs poorly in few-shot classification, where advanced meta-learning algorithms excel; 2) a simple yet effective solution to improve MAML. We note that MAML is almost always cited as a standard but “weak” baseline in few-shot classification, which gives an impression that MAML is simply not suitable for few-shot classification. However, our study shows that MAML, if applied appropriately, can indeed be a fairly strong baseline in few-shot classification, even compared to many recent algorithms. We consider this as an important finding. We certainly know that there are an abundance of algorithms that are specifically designed for few-shot classification, and we are aware of algorithms that are inherently permutation-invariant, as mentioned in the related work of section 6 and at the end of Appendix A. However, we respectfully do not think that this should stop us from digging deeper into a general-purpose algorithm and systematically improving it: none of the previous works have shown that MAML suffers from the permutations. Last but not the least, according to Tables 4, 5, and 6, UNICORN-MAML does outperform ProtoNet consistently, without introducing any extra learnable parameters to the vanilla MAML.
>
> **Good pairing between meta-test task and classifier head.** Thank you for the comment. We will cite [1], and discuss it at the related work of section 6, where we also discussed Proto-MAML and LEO that use class prototypes to initiate the classifiers. One fundamental difference between our work and theirs is in motivation: [1], LEO, and Proto-MAML aim to provide the initialization of the classifier head with better semantic meanings, while our goal is to resolve the sensitivity of MAML to the permutation. We thus respectfully do not consider our work as a “simpler” version of [1]: setting all the classifiers’ initialized weights to be equal can hardly be considered as a good pairing, but an effective way to resolve the permutation issue.
>
> **Motivation.** Thank you for your comment. We apologize for the confusion: our motivation has little to do with the chance of seeing the same set of five classes multiple times during meta-testing. What we really want to solve is that once a set of five classes are sampled, their assignments to {1, 2, 3, 4, 5} have a huge impact on the resulting meta-testing accuracy of that task for MAML (as shown in Section 4). When we gave the motivating example in the introduction (second paragraph on Page 2), some of “them” means some of the “permutations” of a meta-testing task, not some of the “meta-testing tasks”. We will modify our text to clarify this.
>
> **Figure 3.** We thank the question. We use the meta-validation performance to select M, as mentioned in the paragraph below Figure 3: *“Specifically, we first perform meta-training using the M value selected by meta-validation, for each pair of dataset and backbone.”* With this meta-trained MAML model, we then meta-test it (i.e., without outer loop updates) on few-shot tasks sampled either from the meta-training set or meta-testing set. Only in this final meta-testing stage, we consider different numbers of inner loop updates to generate Figure 3.
>
> **What is the point of searching for the best permutation for a meta-training task?** Throughout the whole section 5, we do not make any change to the meta-training phase, as mentioned at the beginning of the section. When we use the support sets’ data as a proxy, the support sets are from the meta-testing tasks, not meta-training tasks.
>
> **About [2].** Thank you for the reference and the suggestion. We will modify our text to claim UNICORN-MAML as a strong baseline for future works.
>
> **Why does UNICORN-MAML work?** Thanks for the comment. We plan to modify this paragraph as follows. The design of UNICORN-MAML ensures that, without using the support set to update the model, the model simply performs at the chance level on the query set. In other words, its formulation, besides permutation-invariant, inherently helps prevent memorization over-fitting (Yin et al., 2020), as it ensures that no single model can solve all tasks at once without inner loop optimization.

---

> ### Comment · Reviewer_eimD · 2021-11-27
> **Thank you authors for the answers. There are still concerns.**
>
> I thank authors for the answers, especially patiently responding to my long list of questions.
>
>
> Another reading the answers and other reviewers’ comments, I still have some concerns. I really appreciate if authors can provide their viewpoints so that I can thoroughly assess this paper and update my rating if appropriate.
>
>
> You have **15 images per class** in in your query set. Therefore, your "huge variance" in Figure 4 is actually just **one more correctly classified sample per class** by cherry-picking the best permuted model from 120 models.  Is it just because of **randomness**, especially with limited sample/update under meta-testing and these models are quite over-fitted?
>
>
> Therefore, the issue of label permutation is just because during meta-testing, classifiers with different final layer weights are used. Maybe meta-testing procedure should be improved to take into account such randomness, but **the issue has nothing specific to intrinsic ideas of MAML**. What is authors’ viewpoint?

---

> > ### Author Response · Authors · 2021-11-29
> > **Thank you for the additional question. Please see our response.**
> >
> > We thank the reviewer for reading our paper and rebuttal. We are pleased to know that our rebuttal addresses the reviewer’s initial concerns. We also thank the reviewer for the additional question and comments. The comment on the query set size is interesting. (We note that the setup with 15 query images per class is quite standard in the literature.) We provide our further response as follows.
> >
> > First, while there exists randomness in creating the query set, we want to emphasize that it is NOT the main cause of the huge variance in Figure 4. To show this, we redo the same experiment as in Section 4, but this time with 500 images per class in the query set. We observe similar variances as in the current manuscript: on one-shot tasks, the best permutation per task leads to a 14% average absolute gain against the worst permutation in TieredImageNet (10% in MiniImageNet); on five-shot tasks, the gain is 6% in TieredImageNet (6% in MiniImageNet). These variances are quite large --- larger than the gap among different algorithms in Table 4 --- and they are caused by the randomness/permutation in class label assignment (as algorithms like ProtoNet won’t have the variance). Also, the best permutation does not merely correctly classify one more example per class, but 10~14% more examples per class on one-shot tasks and 6% more examples per class on five-shot tasks.
> >
> > Second, we want to emphasize that throughout the paper, we do NOT attribute the issue of label permutation to the "intrinsic idea" of MAML, which is to learn the model initialization for fast updates. What we attribute the issue to is how MAML is applied to few-shot classification; i.e., learning the initialization of an N-way classifier whose weight vectors are different. Of course, one may remove this randomness in label permutation by evaluating MAML with all permutations. However, this does not "practically" reduce the variance when MAML is applied to a meta-test task --- there are exponentially many permutations and it is unrealistic to consider all of them (e.g., by performing the model ensemble). We, therefore, believe that a more practical way is to make MAML permutation-invariant, e.g., by unicorn-MAML.
> >
> > We hope these responses answer your question.

---

> > > ### Comment · Reviewer_eimD · 2021-11-29
> > > **Thank you for response.**
> > >
> > > Thank you. I believe the issue of label permutation is just because during meta-testing, classifiers with different final layer weights were used. Randomness and overfitting of adapted models on the limited support set samples have given rise to the observed variance. Importantly, it is not specific to MAML (optimized initial parameters for adaptation).

---

> > > > ### Author Response · Authors · 2021-11-30
> > > > **Thank you and please see our further response.**
> > > >
> > > > We thank the reviewer for the quick response. We appreciate your opinions. More importantly, we respectfully think that your opinions have no conflicts with our contributions/claims/rebuttals and the strengths of our paper listed by other reviewers (e.g., by Reviewer mgm6).
> > > >
> > > > First, about “classifiers with different final layer weights were used”, we respectfully think this is indeed the issue we mentioned in the introduction (as the learned initialization of $w_c$ is unlikely the same across different $c$), the motivation of our Section 4 (first paragraph), and the underlying problem we want to resolve in Section 5 (first paragraphs). For example, in the second to the last paragraph on Page 7, we mentioned *“Concretely, MAML is sensitive to the permutations in class assignments because $w_c\neq w_{c'}$ for $c\neq c'$. One method to overcome this is to make $w_c=w_{c'}$ during meta-testing.”* Our Section 6 further shows that making the meta-training and meta-testing phases consistent (i.e., both permutation-invariant) can lead to a further and often significant gain (please compare Table 4 last row to Table 3). That is, handling the issue of label permutation in both meta-training and meta-testing is better than handling it only in meta-testing. We will be happy to make this point clearer in the final version.
> > > >
> > > > Second, about “... the limited support set samples ...”, we respectfully think this is essentially the core problem of few-shot classification. In our humble opinion, any issues that result from the limited support set (e.g., over-fitting) are nontrivial and are the core problems to resolve in few-shot classification. We note that the randomness and variance mentioned by the reviewer cannot be resolved by evaluating with even more query set samples or even more meta-testing tasks. We do expect that when the support set becomes much larger, the issue of label permutation would become smaller. In particular, when the support set is large, one probably does not need to apply MAML or meta-learning algorithms, but directly trains/fine-tunes the model (for multiple epochs) in a conventional supervised learning way.
> > > >
> > > > Finally, about “it is not specific to MAML”, we do expect that our findings would be applicable to other meta-learning algorithms that are not permutation-invariant by nature, and we will be happy to make this point clearer in the final version. At the beginning of this research project, we detailedly compared MAML to other better-performing meta-learning algorithms like ProtoNet on few-shot classification. We hypothesized that 1) the different final layer weights in MAML’s initialized model and 2) their randomness to be paired with different semantic classes are key issues, and we conducted a systematic study to justify our hypothesis. To our knowledge, we are not aware of any other works that systematically study the effect of these key issues on meta-testing accuracy. Thus, while retrospectively, the label permutation issue may not be specific to MAML, we respectfully think that this would not degrade our contribution: a systematic and focused study of such an issue on one of the most popular and general-purpose meta-learning algorithms. Besides, our work also improves MAML from other aspects. Moreover, we show that by putting things together (i.e., making MAML permutation-invariant and adopting a larger number of inner-loop steps), MAML can be a very strong baseline even compared to recent algorithms on few-shot classification.
> > > >
> > > > In light of our clarifications, we would like to ask if you are willing to reconsider your score, and also if there are any new concerns or additional questions we can respond to!

---

### Author Response · Authors · 2021-11-21
**General responses**

We thank the reviewers for their valuable comments. We are pleased that the reviewers found that our paper is “interesting” (Reviewers eimD, mgm6, Qepb), “systematic” (Reviewers mgm6, zwrd), “thorough/detailed” (Reviewers mgm6, Qepb, zwrd), “encourages rethinking” (Reviewer Qepb), and “novel” (Reviewers mgm6, zwrd). Most of the reviewers gave us positive recommendations (Reviewers mgm6, Qepb, zwrd). In the following, we address the concerns/questions raised by reviewers. We will incorporate all of them in the final version.

---

### Decision · Program_Chairs · 2022-01-20

**Decision:**

Accept (Poster)

**Comment:**

Three of four reviewers rated this paper as an 8.
These positive reviewers felt that this paper provided a lot of value through extensive experimentation with MAML in the few-shot setting. It was felt that the detailed analysis of the inner and outer loop of MAML provided a lot of understanding to the reader regarding the behaviour of MAML. The fourth reviewer giving a score of 3 remains concerned about high variance on some experiments. However the strength of ratings from the other reviewers make the AC more than comfortable giving an accept recommendation for this work.